# COVID-19 genetic risk variants are associated with expression of multiple genes in diverse immune cell types

Benjamin J. Schmiedel [1,6], Job Rocha[1,2,6], Cristian Gonzalez-Colin[1,2,6], Sourya Bhattacharyya [1,6],
Ariel Madrigal[1], Christian H. Ottensmeier [1,3], Ferhat Ay [1,4,7], Vivek Chandra [1,7] &
Pandurangan Vijayanand [1,3,5,7 ✉]

Common genetic polymorphisms associated with COVID-19 illness can be utilized for discovering molecular pathways and cell types driving disease pathogenesis. Given the importance of immune cells in the pathogenesis of COVID-19 illness, here we assessed the effects of COVID-19-risk variants on gene expression in a wide range of immune cell types. Transcriptome-wide association study and colocalization analysis revealed putative causal genes and the specific immune cell types where gene expression is most influenced by COVID-19-risk variants. Notable examples include *OAS1* in non-classical monocytes, *DTX1* in B cells, *IL10RB* in NK cells, *CXCR6* in follicular helper T cells, *CCR9* in regulatory T cells and *ARL17A* in $T_H2$ cells. By analysis of transposase accessible chromatin and H3K27ac-based chromatin-interaction maps of immune cell types, we prioritized potentially functional COVID-19-risk variants. Our study highlights the potential of COVID-19 genetic risk variants to impact the function of diverse immune cell types and influence severe disease manifestations.

[1] La Jolla Institute for Immunology, La Jolla, CA, USA. [2] Center for Genomic Sciences, National Autonomous University of Mexico, Cuernavaca, Morelos, Mexico. [3] Liverpool Head and Neck Centre, Institute of Systems, Molecular and Integrative Biology, University of Liverpool, Liverpool, UK. [4] Department of Pediatrics, University of California San Diego, La Jolla, CA, USA. [5] Department of Medicine, University of California San Diego, La Jolla, CA, USA. [6] These authors contributed equally: Benjamin J. Schmiedel, Job Rocha, Cristian Gonzalez-Colin, Sourya Bhattacharyya. [7] These authors jointly supervised this work: Ferhat Ay, Vivek Chandra, Pandurangan Vijayanand. ✉email: vijay@lji.org

The clinical presentation of SARS-CoV-2 infection in humans can range from very mild or no symptoms to severe respiratory failure[1]. Although hyperactivation of various cellular components of the immune system has been observed in patients with severe COVID-19 illness[2,3], the host genetic factors that determine susceptibility to severe COVID-19 illness are not well understood. Genome-wide association studies (GWAS) addressing this question have identified a number of genetic variants associated with COVID-19 susceptibility and severity[4–8]. Several target genes associated with these COVID-19 risk variants were defined based on their proximity to the risk loci[4–8], although this approach does not accurately prioritize causal genes[9]. To determine putative causal genes, a recent COVID-19 GWAS additionally conducted Mendelian randomization (MR) and transcriptome-wide association study (TWAS), utilizing information from expression quantitative trait loci (eQTLs) datasets in lung tissues and whole blood (GTEx project), and identified significant associations with the expression of seven genes (IFNAR2, TYK2 from MR analysis, and CCR2, CCR3, CXCR6, MAT2B, OAS3 from TWAS analysis)[7].

Because the effects of common genetic variants on gene expression are highly cell-type-specific[10], utilizing eQTL datasets from tissue samples with substantial cellular heterogeneity, like whole blood, is likely to miss associations that are cell-type-specific, and also fail to pin-point the precise cell types where the effects of COVID-19-risk variants are most prominent. Furthermore, the impact of COVID-19-risk variants on gene expression in cell types that play a key role in COVID-19 pathogenesis, such as immune cell types, has not been fully explored. The DICE database of immune cell gene expression, epigenomics, and expression quantitative trait loci (eQTLs) (http://dice-database.org) was established to precisely address these questions as well as to help narrow down functional variants in dense haploblocks linked to disease susceptibility[10,11]. Here, we utilize eQTLs (DICE database) and chromatin accessibility profiles of 13 different immune cell types and two cell types in activated conditions, as well as 3D cis-interactome maps to prioritize target genes and cell types where expression of genes is most affected by genetic variants linked to COVID-19 severity and susceptibility.

## Results

**Target genes associated with COVID-19 risk variants in immune cell types.** We utilized publicly available data from meta-analyses of GWAS for three COVID-19 phenotypes: (A) critical COVID-19 illness, (B) moderate to severe COVID-19 illness requiring hospitalization, and (C) reported SARS-CoV-2 infection (COVID-19 Host Genetics Initiative, release 5 from January 18, 2021[4]; GWAS association P value <5×10^−8; Supplementary Fig. 1a). These meta-analyses reported 19 independent loci that were significantly associated with COVID-19 disease severity (A or B) or susceptibility (C)[12]. To predict cell types that are likely to be major contributors to the genetic risk of COVID-19, we first assessed enrichment of COVID-19-risk variants in cis-regulatory regions from a wide range of cell types and tissues using GARFIELD[13] software. As expected, COVID-19-risk variants showed significant enrichment in chromatin accessibility regions (ATAC-seq and DNase-seq peaks) from blood cells and immune cell types, but little or no enrichment in other tissues and cell types (Supplementary Fig. 1b and Supplementary Data 1).

To determine potential target genes and immune cell types where the effects of COVID-19-risk variants are most prominent, we first assessed the overlap of COVID-19-risk variants with DICE eQTLs from 13 different immune cell types and two cell types in activated conditions. We found that variants in 10

COVID-19-risk loci were significantly associated in cis with the expression of 37 genes (called as eGenes) in immune cell types (Table 1, Supplementary Data 2, and Supplementary Fig. 2). However, due to the high enrichment of GWAS loci in eQTL regions, this simple overlap approach can lead to false associations[9]. Therefore, we utilized two complementary statistical frameworks, colocalization analysis[14,15], and TWAS[9,16,17], to prioritize putative casual genes in immune cell types that are linked to COVID-19 severity or susceptibility. We performed colocalization analysis using the COLOC framework[14,15] to determine the posterior probability of a variant to be significantly associated with both COVID-19 risk and gene expression (eQTL) (see Methods). GWAS variants in 5 independent COVID-19-risk loci showed the high posterior probability of colocalization (PP4 > 0.5 and PP4/PP3 ratio ≥5)[15,18–21] with eQTLs associated with the expression of 9 eGenes (Table 1, Fig. 1a, Supplementary Fig. 3a, b, and Supplementary Data 3). Notably, a large fraction of these eGenes showed prominent effects in specific immune cell types (Fig. 1b). Next, we employed PrediXcan framework[17] to build models, which can predict gene expression in immune cell types using DICE-eQTL dataset. Single-tissue (S-PrediXcan[17]) and integrated (S-MulTiXcan[16]) TWAS using these models showed that the expression of nine genes in four independent loci was significantly associated with COVID-19 severity or susceptibility (Table 1, Fig. 1c, d, Supplementary Fig. 3c, d, and Supplementary Data 4 and 5). For many genes (e.g., OAS1, IL10RB, DTX1, DONSON, CCR9, FYCO1, ACBD4, ARL17A, LRRC37A2, NSF, NXPE3), our TWAS and colocalization analysis using immune cell eQTLs (DICE dataset) provide the first evidence that their expression in specific immune cell types is likely to be causally linked to COVID-19 severity or susceptibility (Table 1).

**The COVID-19 risk allele is associated with reduced OAS1 expression non-classical monocytes.** Potential defects in the type 1 interferon signaling pathway have been reported in patients with severe COVID-19 illness[22–25]. Here, we found that colocalized severe COVID-19-risk variants in chromosome 21 were associated with reduced expression of the gene encoding interferon receptor 2 (IFNAR2) in 12 of the 13 immune cell types analyzed (Fig. 2a). TWAS also showed a significant association of COVID-19 severity with reduced expression of IFNAR2 in multiple immune cell types, thus supporting a role for impaired interferon signaling in immune cells in the pathogenesis of severe COVID-19 illness (Fig. 1c, d). In addition, we found that colocalized severe COVID-19-risk variants in another independent locus (chromosome 12) were also associated with reduced expression of two interferon-inducible genes (OAS1 and OAS3) (Fig. 2a–e). OAS1 and OAS3 encode for the oligoadenylate synthase family of proteins that degrade viral RNA and activate antiviral responses[26]. OAS1 showed a peak COVID-19-risk eQTL (rs4767032, adj. association P value = 1.05 × 10^−6) specifically in patrolling non-classical monocytes (NCM) (Fig. 2a), whereas OAS3 showed prominent eQTLs in T-cell subsets (Fig. 2a), highlighting cell type-restricted effects of COVID-19-risk variants. In NCM, TWAS confirmed that reduced expression of OAS1 is associated with the severity of COVID-19 illness (Fig. 1c, d and Supplementary Data 4 and 5). NCM plays a protective role in viral infections[27], and they have been shown to activate T cells and NK cells as well as produce cytokines that can promote T_H1 immune responses[28–31]. Interestingly, immuno-profiling studies in COVID-19 patients have reported a dramatic reduction in the frequency of NCM in COVID-19 illness[32,33]. Thus, genetic variants that reduce OAS1 expression are likely to impair the degradation of SARS-CoV-2 RNA and antiviral responses

**Table 1 Prioritization of target genes associated with COVID-19-risk GWAS variants in immune cell types.**

| Location of COVID-19 risk locus | Count of significant GWAS SNPs in locus | Most significant GWAS SNP (rsID) | eGenes associated with COVID-19 GWAS variants identified by different methods in DICE dataset | | | | eGenes identified by Pairo-Castineira et al.[7] | Suggested COVID-19 phenotype |
|---|---|---|---|---|---|---|---|---|
| | | | Overlap of COVID-19-risk GWAS variants with DICE eQTLs | Colocalization analysis (COLOC) | TWAS analysis (S-PrediXcan) | TWAS analysis (S-MultiXcan) | | |
| chr1: 155,105,882–155,173,527 | 4 | rs4971066 | RP11-307C12.11, THBS3, GBAP1, MTX1P1, ASH1L | | | | | Hospitalization, infection risk |
| chr3: 45,637,109–46,573,675 | 630 | rs35081325 | CXCR6, FYCO1, CCR1, CCR2 | CXCR6, FYCO1 | CCR9, CXCR6 | CCR9, CXCR6 | CXCR6, CCR3, CCR2 | Critical illness, hospitalization, infection risk |
| chr3: 101,274,911–101,535,347 | 133 | rs10936744 | FAM172BP, ZBTB11-AS1, NXPE3 | NXPE3 | | | | Infection risk |
| chr3: 197,127,009 | 1 | rs9816840 | | | | | | Critical illness |
| chr6: 31,120,729–31,121,958 | 5 | rs111837807 | TCF19 | | | | | Critical illness, hospitalization |
| chr6: 33,055,355 | 1 | rs9501257 | HLA-DMA, RPL32P1 | | | | | Critical illness |
| chr6: 41,502,683 | 1 | rs1886814 | | | | | | Hospitalization |
| chr7: 54,647,894 | 1 | rs622568 | | | | | | Critical illness, hospitalization |
| chr7: 107,607,902 | 1 | rs2237698 | | | | | | Critical illness |
| chr8: 125,336,564 | 1 | rs72711165 | | | | | | Hospitalization |
| chr9: 136,132,012–136,296,530 | 73 | rs8176719 | | | | | | Hospitalization, infection risk |
| chr12: 103,014,757 | 1 | rs10860891 | | | | | | Critical illness, hospitalization |
| chr12: 113,350,796–113,425,679 | 133 | rs10735079 | OAS1, OAS3, DTX1 | OAS1, OAS3, DTX1 | OAS1 | OAS1 | OAS3 | Critical illness, hospitalization, infection risk |
| chr15: 100,774,021 | 1 | rs143434383 | | | | | | Infection risk |
| chr17: 43,705,356–44,863,133 | 1110 | rs1819040 | ACBD4, HEXIM1, SPATA32, LRRC37A4P, DND1P1, CRHR1-IT1, RP11-669E14.6, KANSL1, KANSL1-AS1, ARL17B, LRRC37A2, ARL17A, NSF, WNT3 | NSF | ACBD4,LRRC37A2, ARL17A | LRRC37A2 | | Hospitalization |
| chr17: 47,868,921–47,940,666 | 6 | rs77534576 | ACSF2 | | | | | Critical illness |
| chr19:4,684,642–4,724,734 | 11 | rs2109069 | | | | | | Critical illness, hospitalization, infection risk |
| chr19:10,427,721–10,477,067 | 4 | rs74956615 | PDE4A, CTC-215O4.4 | | | | | Critical illness, hospitalization |
| chr21:34,589,811–34,634,878 | 64 | rs13050728 | IFNAR2, IL10RB | IFNAR2, IL10RB | IFNAR2, IL10RB, DONSON | IFNAR2, IL10RB | IFNAR2 | Critical illness, hospitalization, infection risk |

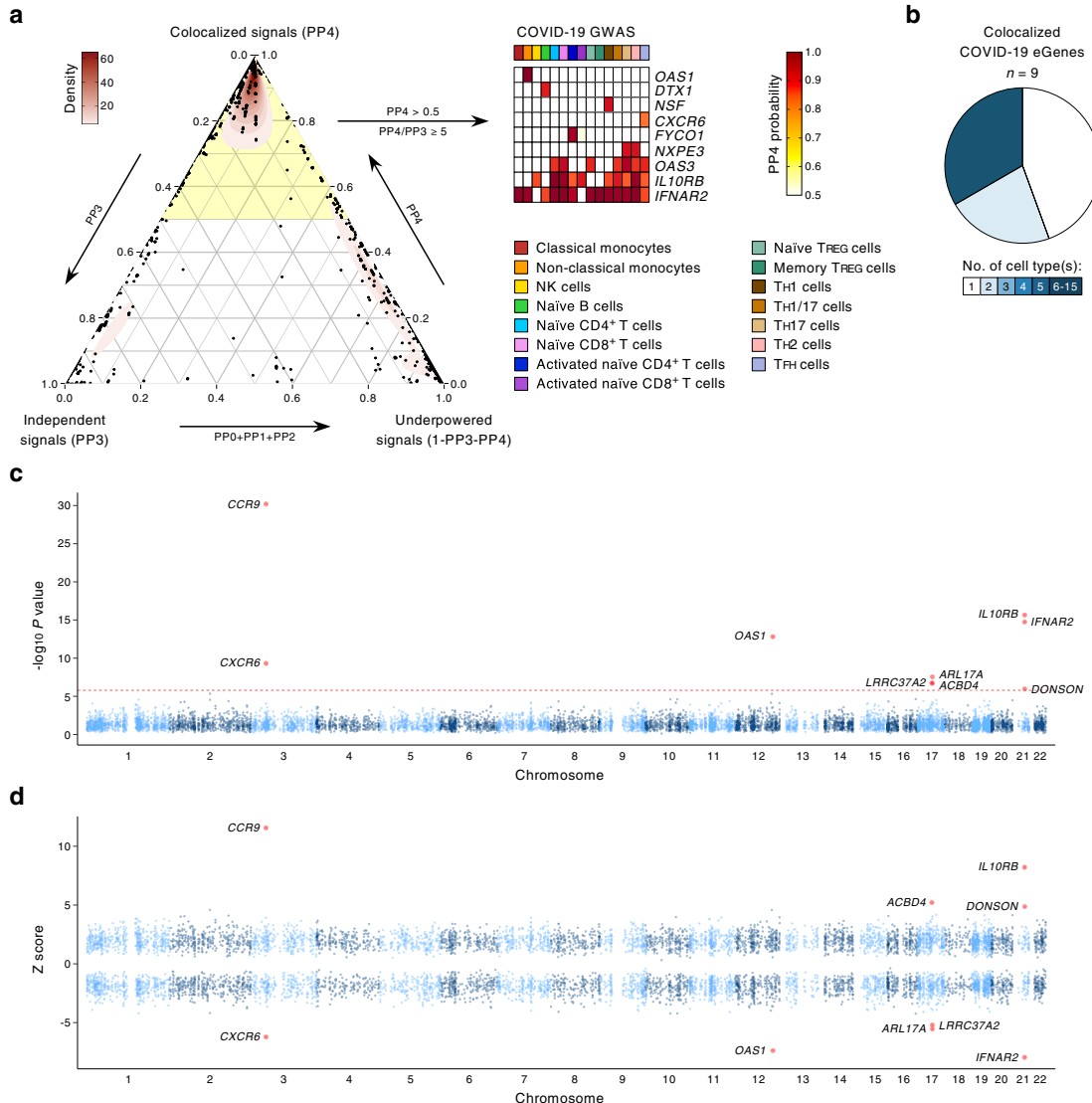

**Fig. 1 Prioritization of genes and immune cell types linked to COVID-19-risk variants. a** Colocalization analysis. Left panel, a ternary plot showing the posterior probabilities of various configurations from COLOC analysis for each gene–cell-type pairs (PP0 = variant with no association, PP1 = variant with association only with COVID-19 phenotype, PP2 = variant with association only with gene expression (PP0 + PP1 + PP2 = underpowered signals), PP3 = independent association of variants with COVID-19 phenotype and gene expression, PP4 = colocalized variant, where a single causal SNPs is associated with both COVID-19 phenotype and gene expression; see "Methods"). Genes with a high probability of colocalized signals (PP4 > 0.5 and PP4/PP3 ratio ≥5, highlighted in the yellow shade and shown in the right panel) in a specific immune cell type represent putative causal genes linked to COVID-19 risk. Right panel, genes, and immune cell types influenced by variants associated with COVID-19 phenotypes (combined results of all three COVID-19 phenotypes from GWAS meta-analyses; see "Methods"). For eGenes with colocalized signals (rows), the posterior probabilities of colocalized GWAS and eQTL signals (PP4) is shown for different immune cell types (columns). **b** Fractions of COVID-19-risk eGenes with colocalized signals identified in varying numbers of immune cell types. **c** Single-tissue TWAS (S-PrediXcan[17]): gene-level Manhattan plot showing the association $P$ value ($-\log_{10}$) for gene expression with COVID-19 phenotypes; results from single tissue TWAS analysis across tissues is shown (see Methods). The red horizontal line shows gene-level Bonferroni corrected genome-wide significant $P$ value threshold ($P < 1.613 \times 10^{-6}$). **d** $Z$ scores showing the direction of effect for the genotype-inferred expression of transcripts that encode protein-coding genes in human immune cell types. Red circles indicate genes with Bonferroni corrected genome-wide significant $P$ value threshold ($P < 1.613 \times 10^{-6}$).

triggered by NCM, which may contribute to the pathogenesis of severe COVID-19 illness.

To explore the molecular mechanisms that explain cell-specific effects of COVID-19-risk variants in the *OAS1* loci, we performed (assay for transposase-accessible chromatin) ATAC-seq analysis of the 13 DICE immune cell types and two activation conditions, and also generated H3K27ac ChIP-seq as well as HiChIP-based chromatin-interaction maps in NCM (see "Methods"). Although the *OAS1* promoter did not directly overlap COVID-19-risk variants, we found several fine-mapped[34,35] COVID-19-risk

variants, associated with *OAS1* expression (eQTLs) (Supplementary Data 6), directly overlapped an intergenic transposase-accessible and H3K27ac-enriched peak region, located 20 kb away from *OAS1* promoter (Fig. 2d). This potential *cis*-regulatory region directly interacted with *OAS1* promoter in NCM (Fig. 2d), which suggested that perturbation of its activity by *OAS1* eQTLs is likely to explain their cell-type-specific effects. The *OAS1* eQTLs in the NCM-specific H3K27ac peak region were predicted to disrupt the binding sites of several transcription factors (Supplementary Data 7). Most notable was the perturbation of

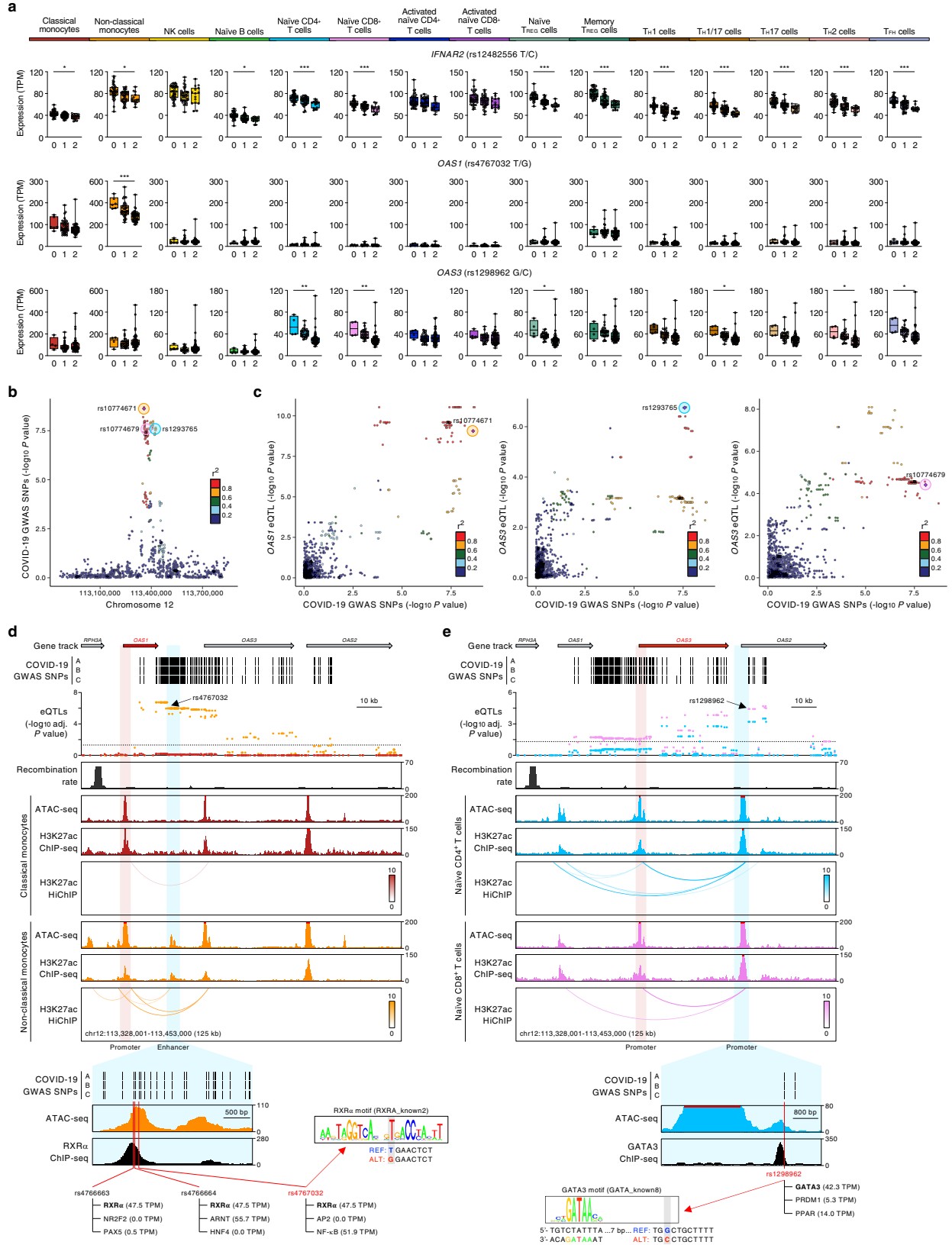

three independent binding sites for retinoid X receptor alpha (RXRα), which has also been shown to bind to this DNA region (ENCODE Transcription factor ChIP-seq in liver tissue and liver cell lines[36]) (Fig. 2d, bottom panel). Given the recognized role of RXRα in regulating gene expression in monocytes[37], we speculate that perturbation of RXRα binding to the NCM-specific *cis*-regulatory region could result in reduced *OAS1* expression specifically in NCM.

**Cell type-restricted effects of COVID-19-risk variants.** The COVID-19-risk variants in the *OAS1*/*OAS3*/*OAS2* locus were also

**Fig. 2 COVID-19-risk variants affect genes in the IFN response pathway. a** Mean expression levels (TPM) of selected severe COVID-19-risk-associated eGenes (all with GWAS association $P$ value $<5 \times 10^{-8}$) identified by colocalization analysis in the indicated cell types from subjects ($n = 91$) categorized based on the genotype at the indicated GWAS cis-eQTL; each symbol represents an individual subject; adj. association $P$ value *$P < 0.05$, **$P < 0.001$, and ***$P < 0.00001$. The boxplots show the 25th percentile, median, and 75th percentile, with the whiskers indicating the minimum and maximum values. **b** Plot shows GWAS association $P$ value for COVID-19-risk variants in the indicated locus; the colocalized variants for *OAS1* in non-classical monocytes (rs10774671, orange circle), *OAS3* in naive CD4+ T cells (rs1293765, blue circle) and naive CD8+ T cells (rs10774679, pink circle) are highlighted. **c** Locus compare plots[63] for *OAS1* in non-classical monocytes (left panel), *OAS3* in naive CD4+ T cells (middle panel), and in naive CD8+ T cells (right panel). Each variant is plotted to represent its $-\log_{10}$ COVID-19 GWAS association $P$ value ($x$ axis) and the $-\log_{10}$ association $P$ value for cis-eQTLs associated with expression of the indicated gene ($y$ axis). The colors denote the strength of linkage disequilibrium (LD) with the indicated lead SNP from the colocalization analysis (purple diamond). **d** Top panel, WashU Epigenome Browser tracks for the *OAS1* locus, COVID-19-risk-associated GWAS variants (based on respective GWAS meta-analyses, see Supplementary Fig. 1a), adj. association $P$ values for cis-eQTLs associated with expression of *OAS1* in classical monocytes and non-classical monocytes, recombination rate tracks[64,65], ATAC-seq tracks, H3K27ac ChIP-seq tracks, and H3K27ac HiChIP-based chromatin-interactions in classical monocytes and non-classical monocytes (NCM). Bottom panel, *OAS1* eQTLs that overlap the NCM-specific cis-regulatory region (chr12:113,362,201-113,365,200; see "Methods"), of which three SNPs at the peak of the transposase-accessible region are predicted to disrupt three distinct binding motifs of RXRα (see Supplementary Data 7) and overlap RXRα binding sites in ENCODE chromatin immunoprecipitation sequencing (ChIP-seq) data of liver tissue and liver cell lines[36]. **e** Top panel, WashU Epigenome Browser tracks for the *OAS3* locus, COVID-19-risk-associated GWAS variants (based on respective GWAS meta-analyses, see Supplementary Fig. 1a), adj. association $P$ values for cis-eQTLs associated with expression of *OAS3* in naive CD4+ T cells and naive CD8+ T cells, recombination rate tracks[64,65], ATAC-seq tracks, H3K27ac ChIP-seq tracks, and H3K27ac HiChIP-based chromatin-interactions in naive CD4+ T cells and naive CD8+ T cells. Bottom panel, *OAS3* eQTLs that overlap *OAS2* promoter region (chr12:113,415,001-113,420,000; see "Methods"), of which one eQTL (rs1298962) is predicted to disrupt the binding of GATA3 (see Supplementary Data 7) and overlaps a GATA3-binding site identified in human thymocytes[39].

associated with reduced expression of *OAS3* in certain T cell subsets but not in monocytes, NK cells, or B cells (Fig. 2a). Accordingly, in naive T cells but not monocytes, the *OAS3* promoter interacted with the promoter of the neighboring gene *OAS2* that harbors *OAS3* eQTLs (peak eQTL rs1298962, adj. association $P$ value $= 3.75 \times 10^{-5}$) (Fig. 2e), suggesting that perturbation of this T cell-specific cis-regulatory interaction is likely to explain the cell-specific effects of *OAS3* eQTLs. This notion is supported by recent reports showing that promoters can interact with neighboring gene promoters and regulate their expression[11,38]. Notably, the *OAS3* eQTL (rs1298962) was also identified as a putative causal variant by fine-mapping[34,35] (Supplementary Data 6) and predicted to disrupt the binding site of GATA3, a T-cell transcription factor that has previously been shown to bind to the neighboring *OAS2* promoter region in human thymocytes[39], suggesting a potentially important role in regulating *OAS3* expression (Fig. 2e, bottom panel and Supplementary Data 7). Of note, we also identified a canonical GATA motif 10 nucleotides upstream of rs1298962 (Fig. 2e, bottom panel), which led us to hypothesize that this variant may modulate the binding affinity of GATA3 either by perturbation of a submaximal recognition motif that affects enhancer syntax ("suboptimization") or by motif independent mechanisms[40–42].

Interestingly, we found that another colocalized severe COVID-19-risk variant (rs2010604, adj. association $P$ value $= 4.50 \times 10^{-2}$) in the *OAS1*/*OAS3*/*OAS2* locus influenced the expression of a neighboring gene (*DTX1*) specifically in naive B cells (Fig. 3a–c). Active chromatin-interaction maps in naive B cells showed that a cis-regulatory region near the eQTL (rs2010604) indirectly interacted with the promoter of *DTX1*, located >80 kb away, and likely modulates its transcriptional activity (Fig. 3d). *DTX1*, encodes for a ubiquitin ligase Deltex1 that regulates NOTCH activity in B cells[43]. Deltex1 has also been shown to promote anergy, a functionally hypo-responsive state, in T cells[44]; if Deltex1 has similar functions in B cells, then genetic modulation of *DTX1* levels may have a profound impact on the function of B cells in COVID-19 illness. In summary, our search for target genes in the severe COVID-19-risk loci in chromosome 12 revealed three putative causal eGenes that display prominent cell-type-restricted effects—*OAS1* in NCM, *OAS3* in T-cell subsets, and *DTX1* in B cells.

**The COVID-19 risk allele is associated with increased *IL10RB* expression in multiple immune cell types.** We found that severe COVID-19-risk variants in the *IFNAR2* locus (chromosome 21) were colocalized with eQTLs linked to increased expression of the neighboring gene *IL10RB* in NK cells and T-cell subsets (rs12482556, adj. association $P$ value $= 4.93 \times 10^{-5}$ in NK cells) (Fig. 3a, e, f). TWAS also confirmed the association of COVID-19 severity with increased expression of *IL10RB* (Fig. 1c, d). H3K27ac HiChIP-based chromatin-interaction maps in NK cells showed that multiple cis-regulatory regions, including the *IFNAR2* promoter, harbor COVID-19-risk eQTLs and directly or indirectly interact with the *IL10RB* promoter and potentially influence its expression (Fig. 3g). *IL10RB* encodes for IL-10 receptor beta, and given the immunomodulatory role of IL-10[45,46], it is likely that the higher expression on the *IL10RB* in NK cells and T cells may enhance their responsiveness to IL-10. Thus, our findings point to a potentially important role for IL-10 signaling and NK cells in influencing the severity of COVID-19 illness.

Genetic variants in the 3p21.31 locus have been linked to the severity of COVID-19 illness by multiple GWAS studies[4–7,12]. The severe COVID-19-risk variants in the 3p21.31 locus contains 17 known protein-coding genes (Fig. 1c), including *SLC6A20*, *LZTFL1*, *CCR9*, *FYCO1*, *CXCR6*, *XCR1*, *CCR1*, *CCR3*, *CCR2*, and *CCR5*. A previous TWAS using whole blood and lung eQTL datasets reported a significant association between COVID-19 severity and expression of chemokine receptor encoding genes *CCR2*, *CCR3*, and *CXCR6* in lung tissue[7]. We found that variants in the 3p21.31 locus colocalized with distinct eQTLs linked to *CXCR6* and *FYCO1* expression in T$_{FH}$ cells and stimulated naive T cells, respectively (Fig. 4a, c). In naive regulatory T cells, TWAS found a significant association between COVID-19 severity and *CCR9*, which encodes for the gut-homing chemokine receptor CCR9[47] (Fig. 1c, d and Supplementary Data 4 and 5). In other T-cell subsets like T$_{FH}$, T$_H$2, and T$_H$1/17 cells, TWAS also revealed associations between expression of genes in chromosome 17 (*ACBD4*, *ARL17A*, and *LRRC37A2*) and COVID-19 disease severity (Fig. 4b, c). As expected, we found the expression of these genes in the corresponding T-cell subset was significantly associated with distinct COVID-19-risk variants in this region.

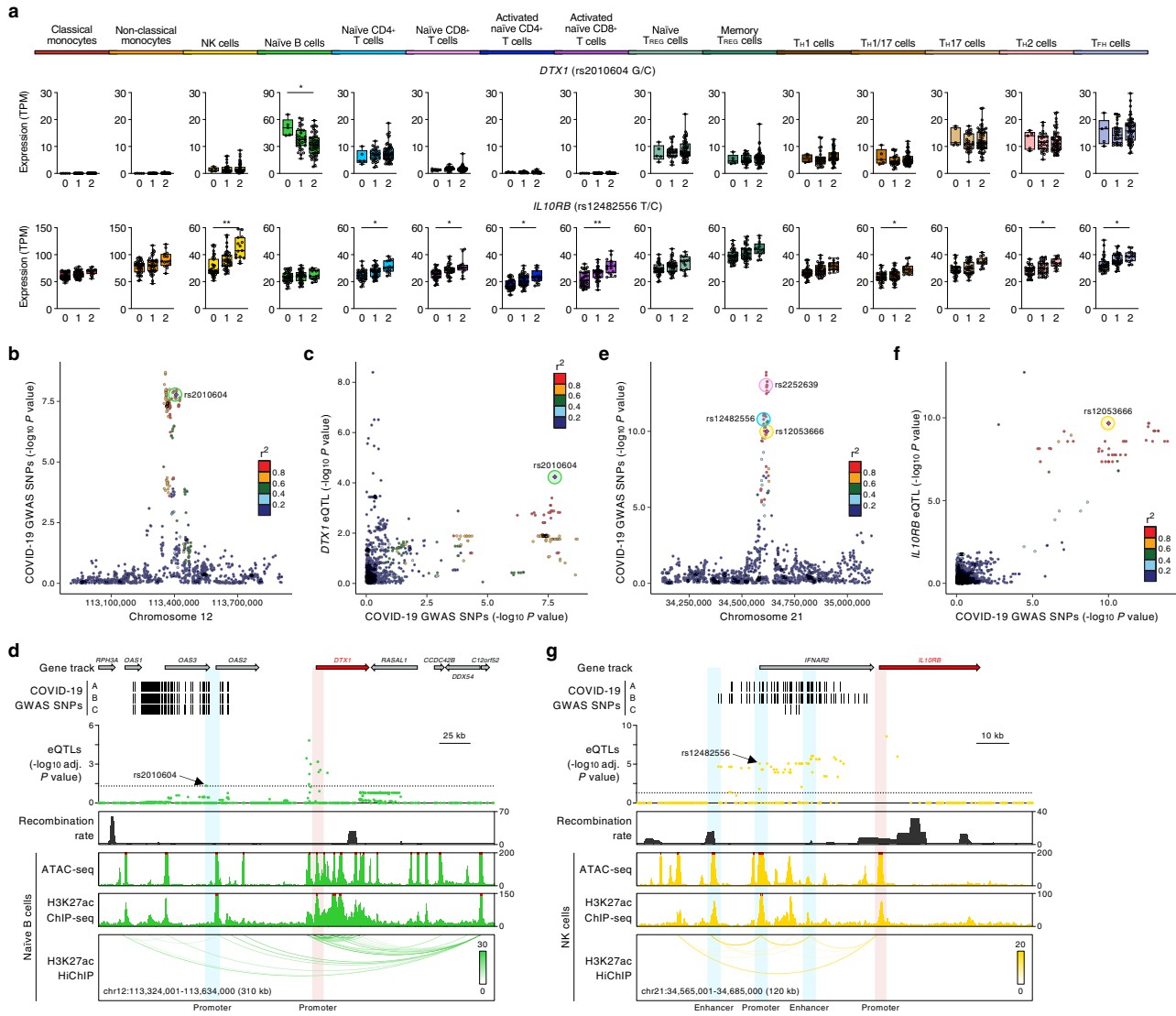

**Fig. 3 Cell type-restricted effects of COVID-19-risk variants. a** Mean expression levels (TPM) of severe COVID-19-risk-associated GWAS eGenes *DTX1* and *IL10RB* in the indicated cell types from subjects ($n = 91$) categorized based on the genotype at the indicated GWAS *cis*-eQTL; each symbol represents an individual subject; adj. association *P* value *$P < 0.05$, **$P < 0.001$, and ***$P < 0.00001$. The boxplots show the 25th percentile, median, and 75th percentile, with the whiskers indicating the minimum and maximum values. **b** Plot shows GWAS association *P* value for COVID-19-risk variants in the indicated locus; the colocalized variant for *DTX1* in naive B cells (rs2010604, green circle) is highlighted. **c** Locus compares plots[63] for *DTX1* in naive B cells. Each variant is plotted to represent its -$\log_{10}$ COVID-19 GWAS association *P* value (x axis) and the −$\log_{10}$ association *P* value for *cis*-eQTLs associated with the expression of *DTX1* (y axis). The colors denote the strength of linkage disequilibrium (LD) with the indicated lead SNP from the colocalization analysis (purple diamond). **d** WashU Epigenome Browser tracks for the *DTX1* locus, COVID-19-risk-associated GWAS variants (based on respective GWAS meta-analyses, see Supplementary Fig. 1a), adj. association *P* values for *cis*-eQTLs associated with expression of *DTX1* in naive B cells, recombination rate tracks[64,65], ATAC-seq tracks, H3K27ac ChIP-seq tracks, and H3K27ac HiChIP-based chromatin-interactions in naive B cells. **e** Plot shows GWAS association *P* value for COVID-19-risk variants in the indicated locus; the colocalized variants for *IL10RB* in NK cells (rs12053666, yellow circle), and *IFNAR2* in naive CD4+ T cells (rs12482556, blue circle) and naive CD8+ T cells (rs2252639, pink circle) are highlighted. **f** Locus compares plots[63] for *IL10RB* in NK cells. Each variant is plotted to represent its −$\log_{10}$ COVID-19 GWAS *P* value (x axis) and the −$\log_{10}$ association *P* value for *cis*-eQTLs associated with the expression of *IL10RB* (y axis). The colors denote the strength of linkage disequilibrium (LD) with the indicated lead SNP from the colocalization analysis (purple diamond). **g** WashU Epigenome Browser tracks for the *IL10RB* locus, COVID-19-risk-associated GWAS variants (based on respective GWAS meta-analyses, see Supplementary Fig. 1a), adj. association *P* values for *cis*-eQTLs associated with expression of *IL10RB* in NK cells, recombination rate tracks[64,65], ATAC-seq tracks, H3K27ac ChIP-seq tracks, and H3K27ac HiChIP-based chromatin-interactions in NK cells.

## Discussion

Several COVID-19-risk variants show cell-type-restriction of their effects on gene expression, and thus have the potential to impact the function of diverse immune cell types and gene pathways. Our analysis of eQTLs, transposase accessible chromatin profiles, and *cis*-interaction maps in multiple immune cell types enabled a precise definition of the cell types and genes that drive genetic susceptibility to severe COVID-19 illness, potentially contributing to different clinical outcomes. However, COVID-19-risk variants may display stronger associations with gene expression in other immune cell types and activation conditions not examined in this work. Our study also highlights how information about common genetic polymorphisms can be used to define molecular pathways and cell types that play a role in disease pathogenesis.

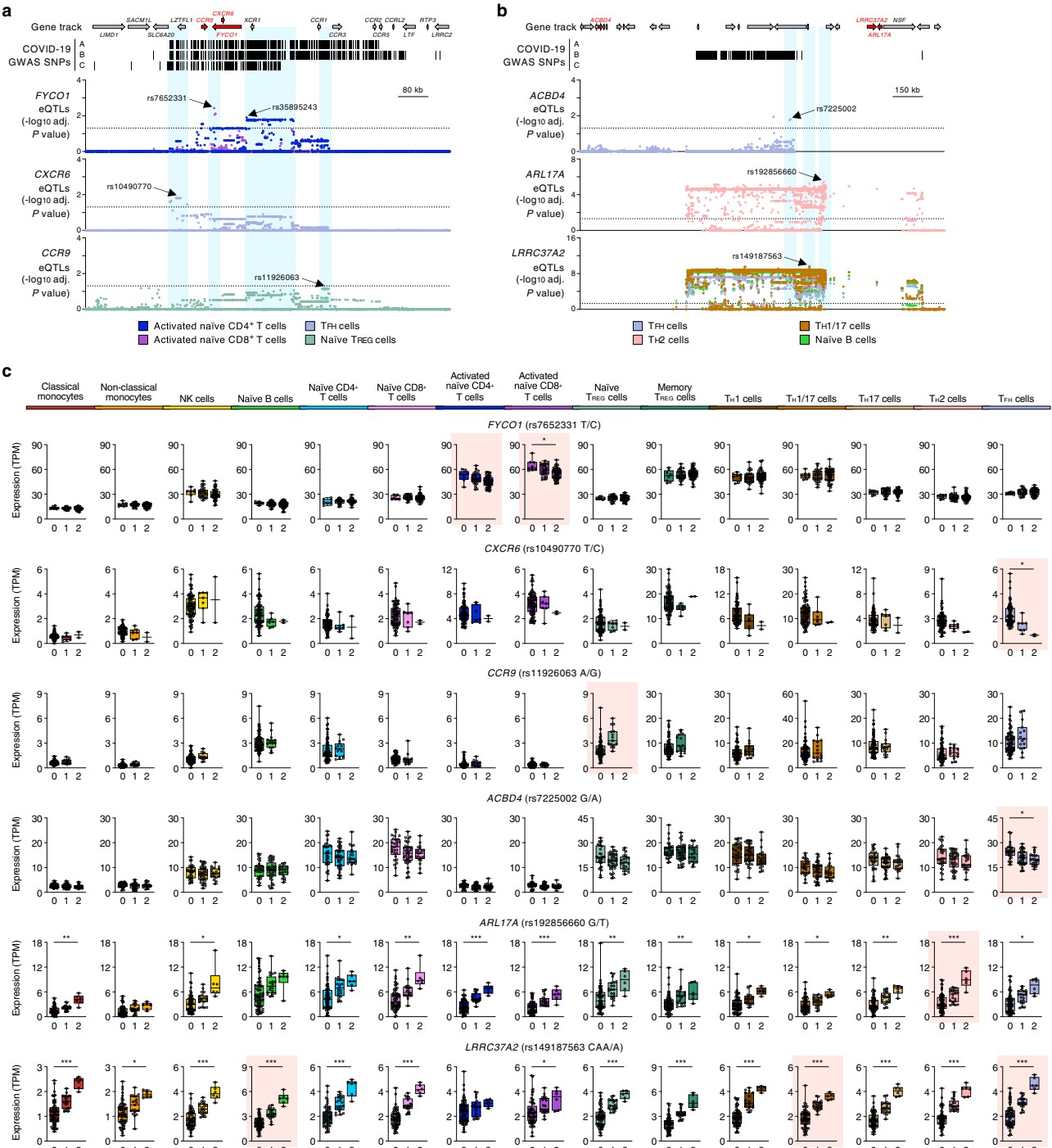

**Fig. 4 COVID-19-risk-associated genes prioritized by colocalization analysis and TWAS. a** WashU Epigenome Browser tracks for the gene locus on chromosome 3 harboring the genes *FYCO1*, *CXCR6*, and *CCR9*, COVID-19-risk-associated GWAS variants (based on respective GWAS meta-analyses, see Supplementary Fig. 1a), adj. association *P* values for *cis*-eQTLs associated with expression of the indicated genes in various immune cell types and conditions. **b** WashU Epigenome Browser tracks for the gene locus on chromosome 17 harboring the genes *ACBD4*, *LRRC37A2*, and *ARL17A*, COVID-19-risk-associated GWAS variants (based on respective GWAS meta-analyses, see Supplementary Fig. 1a), adj. association *P* values for *cis*-eQTLs associated with expression of the indicated genes in various immune cell types and conditions. **c** Mean expression levels (TPM) of selected COVID-19-risk-associated GWAS eGenes (GWAS association *P* value <5 × 10$^{-8}$), in the indicated cell types from subjects (*n* = 91) categorized based on the genotype at the indicated GWAS *cis*-eQTL; each symbol represents an individual subject; adj. association *P* value *P* < 0.05, **P* < 0.001, and ***P* < 0.00001. The boxplots show the 25th percentile, median, and 75th percentile, with the whiskers indicating the minimum and maximum values. Significant associations with COVID-19 GWAS variants found by colocalization (*CXCR6* and *FYCO1*) or TWAS analysis (*CCR9*, *CXCR6*, *ACBD4*, *ARL17A*, and *LRRC37A2*) are highlighted for each specific immune cell type.

## Methods

**Leukapheresis samples**. The Institutional Review Board (IRB) of the La Jolla Institute for Allergy and Immunology (LJI; IRB protocol no. SGE-121-0714) approved the study. For the DICE study, a total of 91 healthy volunteers were recruited in the San Diego area, who provided leukapheresis samples at the San Diego Blood Bank (SDBB) after written informed consent. All study subjects self-reported ethnicity and race details, and were tested negative for hepatitis B, hepatitis C, and human immunodeficiency virus (HIV).

**DICE eQTLs and re-analysis of HLA transcript levels using HLApers**. Details of gene expression and eQTL analysis in 13 immune cell types and two cell types in activated conditions have been reported for the DICE project (recalculated to incorporate four previously missing RNA-seq samples; GENCODE annotation v19 (GRCh37.p13))[10]. HLA genes are highly polymorphic leading to inaccurate quantification of transcript levels from RNA-seq which hinders eQTL identification. To address this problem, we re-analyzed our RNA-seq data for all cell types using a recent method (HLApers[48]) that quantifies HLA transcript levels considering the polymorphic nature of HLA genes. Briefly, we applied the HLApers pipeline (https://github.com/genevol-usp/HLApers) for in silico HLA mapping and obtaining transcript expression as performed earlier for all remaining cell types. This pipeline generates personalized HLA index files for individual samples, which are used to estimate sample-specific HLA genotype. We applied HLApers with its default settings, aside from customizing it to support our single-end reads. We executed HLApers by using STAR as the aligner and Salmon for quantifying the transcripts. Matrix eQTL was used to obtain eQTLs from the HLApers-computed gene expression values for HLA genes. For all downstream analysis, we replaced the HLA eQTLs from the initial DICE release with this revised analysis.

**COVID-19 GWAS datasets and overlap with DICE eQTLs**. Genetic variants associated with COVID-19 phenotypes: (A) critical COVID-19 illness, (B) moderate to severe COVID-19 illness requiring hospitalization and (C) reported SARS-CoV-2 infection, were downloaded from the COVID-19 Host Genetics Initiative (release 5 from January 18, 2021). Genetic variants with GWAS association $P$ value $< 5 \times 10^{-8}$ were utilized for downstream analysis. Linkage disequilibrium (LD) for lead COVID-19-risk variants was calculated using PLINK v1.90b3w[49] for continental "super-populations" (AFR, AMR, EAS, EUR, SAS) based on data from phase 3 of the 1000 Genomes Project[50]. SNPs in tight genetic linkage with GWAS lead SNPs (LD threshold $r^2 > 0.80$) in any of the five super-populations were retrieved along with the SNP information (e.g., genomic location, allelic variant, allele frequencies). Utilizing this dataset, GWAS SNPs (lead SNPs and SNPs in LD) were analyzed for overlap with eQTLs in the DICE database (raw $P$ value <0.0001, adj. association $P$ value (FDR) < 0.05, TPM >1.0) separately for each cell type to identify COVID-19-risk variants that were associated with gene expression in immune cell types (Table 1 and Supplementary Data 2).

**Colocalization analysis**. To determine whether the association of GWAS variant with COVID-19 phenotypes is mediated through regulation of gene expression by the variant (eQTL), we employed colocalization analysis using the COLOC framework[14,15]. COLOC estimates the posterior probability that a GWAS variant and an eQTL in a given COVID-19-risk loci share the same causal variant. First, we extracted significant GWAS variants associated with COVID-19 phenotypes (association $P$ value $<5 \times 10^{-8}$; COVID-19 Host Genetics Initiative; release 5 from January 18, 2021) and obtained GWAS summary statistics: association $P$ values, effect size estimate (β), and standard error of β. For each eGene in immune cell types (DICE eQTL database), we extracted eQTL summary statistics: association $P$ values, the effect size estimate (β), standard error of β, and minor allele frequency for all the variants within 1 Mb of TSS of that eGene. As suggested in a previous report[51], we excluded all variants within the MHC locus (chr6: 28,477,897–33,448,354). For each eGene, both eQTL and GWAS summary statistics were then applied to the *coloc.abf* routine of the COLOC package. For each immune cell type ($n = 15$), an independent COLOC analysis was performed using GWAS summary statistics from each COVID-19 phenotype. We used the default setting (p1 or p2 $= 1 \times 10^{-4}$) for the prior probability of a variant being associated with either COVID-19 phenotype (p1) or gene expression (p2). The prior probability (p12) of a variant to be associated with both COVID-19 phenotype and gene expression was determined by sensitivity analysis, as recommended, and set at $1 \times 10^{-5}$ (see Supplementary Fig. 3a for sensitivity analysis of the *IFNAR2* locus in naive CD8$^+$ T cells). An eGene was said to have evidence of colocalization when the posterior probability of colocalization of a GWAS variant and an eQTL linked to the eGene (PP4) was greater than 0.5 and PP4/PP3 ratio ≥5[15,18–21]. As the COLOC framework assumes a single causal variant per colocalized region, we listed the corresponding colocalized variant in Supplementary Data 3. Finally, we selected only those variants (and corresponding eGenes) which are either eQTLs or exhibit strong LD ($r^2 > 0.80$) with an eQTL. LD was computed using PLINK (v1.90b3w).

**TWAS analysis**. We performed a transcriptome-wide association study (TWAS) using DICE eQTL datasets from 13 immune cell types and two cell types in activated conditions. We used the PrediXcan package[17] with recommended minor modifications[52] to first generate prediction models of gene expression in immune cell types using the DICE eQTL database. We utilized genotype data and normalized gene expression data (transcript per million (TPM)) to perform prediction model training, as described[17]. Briefly, we limited our analysis to genes annotated as protein-coding, lncRNA, or miRNA. The first two principal components of the genotyping data were used as covariates for the analysis. For each gene-cell-type pair, we used the Elastic Net algorithm to train the prediction models. We used default settings in PrediXcan: 0.5 as mixing parameter and use genetic variants in a 1 Mb region upstream of the gene start site and 1 Mb downstream of the gene end site, and then performed tenfold cross-validation. Gene prediction models with a cross-validated correlation value >0.1 and a cross-validated prediction performance $P$ value <0.05 were used for downstream analysis.

We then performed transcriptome-wide association applying the MetaXcan framework[17]. For single-tissue TWAS, the S-PrediXcan framework[17] was utilized for performing a total of 30,997 gene-cell-type pairs (15 immune cell types/ conditions) using our prediction models and GWAS summary statistics for all reported COVID-19 phenotypes. We used a Bonferroni corrected $P$ value threshold ($P < 0.05$/number of gene-cell-type pairs tested (30,997) $= P < 1.613 \times 10^{-6}$) to determine significant associations between genes located near significant GWAS variants (<1 Mb) and COVID-19 phenotypes in each immune cell type.

To minimize multiple testing burdens, we also applied the S-MulTiXcan framework[16], which integrates transcriptome prediction models across all cell types by considering correlation across tissues. S-MulTiXcan was performed using cross-cell-type SNP covariance, transcriptome prediction models and GWAS summary statistics for all reported COVID-19 phenotypes. The cross-cell-type SNP covariance was computed using all variants for every gene across different cell types. To identify significant associations between genes located near significant GWAS variants (<1 Mb) and COVID-19 phenotypes, we used a Bonferroni corrected $P$ value threshold ($P < 0.05$/number of genes in at least one prediction model (10,395) $= P < 4.81 \times 10^{-6}$).

**Omni-ATAC-seq**. Omni-ATAC-seq for immune cell types from two DICE donors in technical duplicates was performed as described previously[53]. Following the protocol and gating strategy previously reported[10], immune cells were enriched from peripheral blood mononuclear cells (PBMC) and sorted by FACS. For each technical replicate, 50,000 sorted cells were pelleted and washed once in phosphate-buffered saline (PBS). Nuclei were isolated with lysis buffer and washed with wash buffer as described[53]. The cells were then resuspended in 50 μl of transposition mixture and incubated at 37 °C for 30 min. After transposition DNA was purified using "DNA Clean & concentrator" kit (Zymo Research). Before library amplification, DNA was pre-amplified for five cycles to determine the required cycles of amplification. PCR amplification of DNA was performed and purified libraries were size-selected to 50–600 bp using AMPure XP beads (Beckman Coulter Life Sciences) according to the manufacturer's protocol and subjected to $2 \times 100$ bp paired-end sequencing on NovaSeq6000 (Illumina), see Supplementary Data 8a for details of sequencing libraries.

**ATAC-seq data analysis**. For analysis of ATAC-seq data, we utilized our custom ATAC-seq data processing pipeline ATACproc (https://github.com/ay-lab/ATACProc). Briefly, single-end ATAC-seq reads were aligned to hg19 reference genome using Bowtie2 (version 2.3.3.1)[54,55], with parameters -k 4 –mm –threads 8 –X 2000. We excluded the reads corresponding to the mitochondrial genome and chromosome Y. Uniquely mapped reads with mapping quality ≥ 30 were retained using SAMtools (version 1.6)[56]. Duplicate reads were discarded by Picard tool's MarkDuplicates routine (https://broadinstitute.github.io/picard). To account for the 9 bp distance between two adapters inserted by the Tn5 transposase[57], we then shifted all the reads aligned to the positive (+) strand by +4 bp, and the reads aligned to the negative (−) strand by −5 bp, using Deeptools alignmentSieve routine[58]. We also discarded reads overlapping with the blacklisted regions (provided in https://github.com/Boyle-Lab/Blacklist/tree/master/lists). Coverage tracks were normalized by scaling factor, reads per kilobase per million mapped reads (RPKM), using the "BamCoverage" routine from deepTools[58] using the arguments "-bs 10—effectiveGenomeSize 2864785220—normalizeUsing RPKM -e 200". MACS2 (version 2.1.0)[59] was used for peak calling, with parameters: "-g hs –q 0.05—nomodel—nolambda—keep-dup all—call-summits—shift -100—extsize 200". WashU Epigenome Browser was used to display the tracks.

**Enrichment of COVID-19-risk variants in chromatin accessibility sites**. We utilized GARFIELD (v2) to investigate the enrichment patterns of COVID-19-risk variants at chromatin accessibility sites, identified by ATAC-seq analysis of the 13 DICE cell types and 2 activation conditions, and pre-defined "peaks" from ENCODE, GENCODE, and Roadmap Epigenomics project (excluding all fetal tissues; $n = 224$ in total)[13]. In brief, GARFIELD tool evaluates enrichment using a logistic regression model that accounts for allele frequency, distance to the TSS of the nearest gene, and the number of LD proxies ($r^2 \geq 0.80$; correlation based on the UK10K dataset provided by the software) to extract a set of independent variants and annotates those to a known regulatory region. Enrichment odds ratios (OR) were calculated at various GWAS $P$ value thresholds (T; data retrieved from GWAS meta-analysis B2_ALL_leave_23andme), and significant enrichment patterns were identified using a Bonferroni corrected $P$ value threshold ($P < 9.8 \times 10^{-5}$).

**Fine-mapping of COVID-19-risk variants**. We employed the FINEMAP package[34,35] with default settings to perform statistical fine-mapping of the COVID-19-risk variants and their LD SNPs. Summary statistics for individual genetic variants such as reference and alternate alleles, MAF, $P$ value, and beta were obtained from the COVID-19 Host Genetics Initiative (association $P$ value $<5 \times 10^{-8}$; data retrieved from each GWAS meta-analysis). In brief, for each GWAS locus, we defined a fine-mapping region as a 3 Mb window around a significant variant ($P$ value $<5 \times 10^{-8}$) and merged overlapping regions (as suggested in https://github.com/FINNGEN/finemapping-pipeline). The individual GWAS meta-analyses were pre-processed into separate files per region, and fine-mapping was applied on the individual files using the package LDstore[60] to compute the LD statistics. FINEMAP was employed by allowing a maximum of ten causal SNPs (—n-causal-snps 10) and executed both stepwise conditioning (—cond) and shotgun stochastic search (—sss). A COVID-19-risk variant was defined as a fine-mapped variant if it overlapped with fine-mapping outputs from either stepwise conditioning or shotgun stochastic search approaches.

**H3K27ac ChIP-seq and HiChIP for non-classical monocytes**. ChIP-seq for H3K27ac modification for non-classical monocytes from 6 DICE subjects were performed as described previously[11]. Briefly, 500,000 FACS-sorted cells were crosslinked with 1% formaldehyde and flash-frozen in liquid nitrogen. Sheared chromatin from each sample was then immunoprecipitated with a polyclonal anti-H3K27ac antibody (C15410196; Diagenode) by use of an automated ancillary liquid handler SX-8G IP-Star from Diagenode. Immunoprecipitated chromatin was captured, washed, Illumina library adaptors integrated by transposase-based method, and library prepared by PCR amplification. Libraries were sequenced on an Illumina HiSeq2500 sequencer to obtain 50-bp single-end reads. ChIP-seq reads were analyzed as described previously[11]. Briefly, reads were aligned using Bowtie2 to hg19 reference genome. De-duplicated aligned reads were merged using SAMtools (http://samtools.sourceforge.net/) for all donors, to produce aggregate ChIP-seq reads. MACS2 was used for peak calling. WashU Epigenome Browser was used to display the tracks.

H3K27ac HiChIP for non-classical monocytes from 6 DICE subjects were performed as described previously with some modifications[11]. Briefly, 1 M FACS-sorted cells were crosslinked with 1% formaldehyde and flash-frozen in liquid nitrogen. The nuclear fraction was isolated from fixed cells, the chromatin was digested in intact nuclei using 100 U of the 4-base cutter MboI (New England Biolabs), and the restricted ends were re-ligated. Pelleted nuclei were dissolved in 70 μl nuclear lysis buffer (50 mM Tris-HCl, pH 7.5, 10 mM EDTA, and 0.5% SDS) and sonicated using a Bioruptor for 6 cycles with 16 sec ON and 32 sec OFF. Sonicated chromatin was diluted 10 times in ChIP Dilution Buffer (50 mM Tris-HCl, pH 8.0, 167 mM NaCl, 1.1 mM EDTA, 0.55 mM EGTA, 0.11% Na-deoxycholate, 1.1% Triton X-100, 0.05% SDS, and 1X protease inhibitors) and immunoprecipitated overnight at 4 °C by incubation with 1.0 μl of H3K27ac antibody (C15410196; Diagenode), pre-coated on 15 μl protein A coated magnetic beads (Thermo Fisher Scientific). Immunocomplexes were captured, washed, and resuspended in 100 μl of elution buffer (50 mM NaHCO$_3$ and 1% SDS) as described[11]. After reverse cross-linking at 65 °C overnight and treatment with proteinase K (Thermo Fisher Scientific), DNA was purified using affinity columns (Zymo Research) according to the manufacturer's protocol. After adapter ligation, the biotinylated DNA was captured using Streptavidin C-1 beads according to the manufacturer's protocol and resuspended in 20 μl of DNA elution buffer (10 mM Tris-HCl). PCR amplification of DNA was performed and purified HiChIP libraries were size selected to 300–800 bp using AMPure XP beads (Beckman Coulter Life Sciences) according to the manufacturer's protocol and subjected to 2 × 150 bp paired-end sequencing on NovaSeq6000 (Illumina), see Supplementary Data 8b for details of sequencing libraries.

**Analysis of HiChIP data**. HiChIP data was analyzed as described previously[11]. Briefly, we applied the HiC-Pro pipeline[61] and mapped each end independently to hg19 reference genome, using the aligner Bowtie2. Aligned reads were then paired, and only paired reads involving two different MboI restriction sites were retained. Valid read pairs of individual samples within the distance range 10 kb–3 Mb were merged and 70 M valid pairs were randomly selected to create one aggregate HiChIP contact map. FitHiChIP[62] was then applied as described previously[11] to call statistically significant loops. A 5 kb fixed-sized bin was used to attribute a bin as peak-bin if that bin overlaps with a ChIP-seq or HiChIP-inferred peak in the reference peaks file. FitHiChIP applies fixed-size binning (here 5 kb) on the input set of valid HiChIP reads and attributes a bin as peak-bin if that bin overlaps with a ChIP-seq or HiChIP-inferred peak in the reference peaks file, subject to 1 bp minimum overlap without any slack. Otherwise, the bin is labeled as a non-peak-bin. The background (set of locus pairs used to infer the null model) as well as the foreground (i.e., set of locus pairs that were assigned a significance estimate) of FitHiChIP can be either peak-to-peak (i.e., interactions between two peak bins) or peak-to-all (i.e., interactions involving peak bins in at least one end). The default mode of FitHiChIP uses peak-to-all pairs for the foreground, which is the setting employed in this study. The binomial distribution is employed on the generated contact probabilities to estimate the $P$ values, which are then corrected for multiple testing. Interactions with an FDR $< 0.01$ are considered significant and reported as loop calls. For the HiChIP samples reported in this study, we executed FitHiChIP using both loose and stringent background models but results reported in this study were from the stringent (S) background model of FitHiChIP[62].

**Analysis to identify transcription factor motifs perturbed by COVID-19-risk-associated variants**. COVID-19-risk variants were prioritized as potentially functional based on their overlap with *cis*-regulatory regions that interacted with target gene promoters (pieQTL) as described earlier[11], and the transcription factor motifs perturbed by these COVID-19-risk variants were obtained from the HaploReg v4.1 database (https://pubs.broadinstitute.org/mammals/haploreg/haploreg.php). To identify transcription factors that bind to intergenic *cis*-regulatory region interacting with *OAS1* promoter, we examined the ENCODE Transcription Factor Binding tracks containing 338 transcription factor ChIP-seq peaks in 130 cell types, and extracted RXRα ChIP-seq data from liver tissue and liver cell lines[36]. To determine GATA3 binding near the *OAS2* gene promoter we utilized the GATA3 ChIP-seq data from human thymocytes[39].

**Statistical analysis and data display**. The number of subjects, samples, and replicates analyzed, and the statistical test performed are indicated in the figure legends. GraphPad Prism 9.1.0 software was used for generating graphs and performing statistical significance tests. ATAC-seq peaks, ChIP-seq peaks, and HiChIP interactions were visualized using WashU Epigenome Browser.

**Reporting summary**. Further information on research design is available in the Nature Research Reporting Summary linked to this article.

## Data availability

The DICE project is providing anonymized data for public access at http://dice-database.org. Individual-specific RNA-sequencing and genotype data, H3K27ac ChIP-seq, and HiChIP data in five common immune cell types have been previously reported[10,11]. H3K27ac ChIP-seq and HiChIP data for non-classical monocytes and ATAC-seq data for 15 DICE cell types were newly generated. All datasets have been deposited in the Database of Genotypes and Phenotypes (dbGaP accession number: phs001703.v4.p1). All relevant data supporting the findings of this study are available from the corresponding author upon reasonable request.

## Code availability

The codes used for colocalization analysis, TWAS analysis, and GWAS overlap analysis are available on Github at https://github.com/vijaybioinfo. The codes used for ATAC-seq and HiChIP data analysis are available on GitHub at https://github.com/ay-lab.

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

## Acknowledgements

This work was funded by NIH grants R24-AI108564 (P.V., F.A., and C.H.O.), the William K. Bowes Jr Foundation (P.V.), and R35-GM128938 (F.A.). Utilized equipment was supported by the NIH grants S10RR027366 (BD FACSAria II), S10OD016262 (Illumina HiSeq2500), and S10OD025052 (Illumina NovaSeq6000).

## Author contributions

B.J.S., V.C., C.H.O., F.A., and P.V. conceived the work. B.J.S and V.C. designed and performed the experiments. B.J.S, F.A., V.C., and P.V. wrote the manuscript. J.R., C.G.-C., S.B., and A.M. performed bioinformatic analyses under the supervision of B.J.S., F.A., V.C., and P.V.

## Competing interests

The authors declare no competing interests.
