## [Peer Review File · Nature Communications]

Title: COVID-19 genetic risk variants are associated with expression of multiple genes in diverse immune cell typesEditorial Note: This manuscript has been previously reviewed at another journal that is not operating a transparent peer review scheme. This document only contains reviewer comments and rebuttal letters for versions considered at *Nature Communications*.

REVIEWER COMMENTS

Reviewer #3 (Remarks to the Author):

Most of the concerns that I raised in my original review have since been addressed and as such the study appears suitable for publication.

However, I first request that the authors make some version of the code they used to perform the analysis publicly available (on github or another repository) as opposed to available upon request. Having access to the exact code used is invaluable for understanding the steps performed for the analysis and reproducing the main findings.

Reviewer #4 (Remarks to the Author):

Schmiedel, Chandra, and Rocha et al. report potential genetic mechanisms mediating COVID-19 illness using an array of genomic data types from diverse immune cell types. In the interest of full disclosure, I previously reviewed a version of this manuscript at Nature Genetics. I do find the manuscript to be quite improved and the authors have made marked effort to address my previous concerns which I've outlined below along with any additional remaining concerns.

Previously, my concern was with the rigor in which the "causative" variants were defined. The authors have addressed this in two primary ways – (1) They have incorporated ATAC-seq data and (2) They have, in some cases, delved further into the underlying mechanisms of the genetic association. This is particularly true with the anecdote shown in Figure 2d. This is very strong and I would say that it is almost certain that the authors have identified the causative allele. This is precisely the type and depth of analysis I was originally hoping to see.

The remaining anecdotes now include additional data but do not rise to the same level of confidence as the one shown in Figure 2d. The anecdote in Figure 2e is less convincing due to the absence of a GATA motif overlapping the SNP of interest. It is nice to see that the GATA3 ChIP-seq signal lines up but the actual sequence change does not reside in or near a canonical GATA factor motif (as far as I can tell). Similarly, the anecdotes presented in Figure 3d and Figure 3g lack specific mechanisms and do not posit alteration of a specific motif. While I feel like this is the current standard in the field to nominate a putative causative noncoding variant, I also acknowledge that it may not be possible to find additional examples like the one presented in Figure 2d. In this case, I think the authors should at least mention this as a caveat and indicate that other cell types or cell states not assayed in this work could better explain the given associations.

Beyond identifying additional high-confidence anecdotes, I have the following minor comments:

The x axis in Figure 2a has no label. The y axis reads “Expression” but this is not sufficiently descriptive. The figure legend says TPM - please label as such. This also applies to the other figures of this type throughout the manuscript.

In Table 1, the authors could add a column indicating their predicted causative variant for the relevant loci.

Reviewer #5 (Remarks to the Author):

In this paper, Schmiedel et al., used the DICE data to investigate how GWAS variants associated with the COVID-19 risk impact gene expression in different immune cell types. I agree what other reviewers have said, and believe that this is still largely a preliminary and descriptive analysis. The novelty of the study is limited, or should be better explained.

As a substitute for reviewer 4, I believe that the comments suggested by a reviewer have been addressed. As suggested, the authors performed both coloc and TWAS which strengthened their conclusions, compared to simple overlaps. The authors also improved their method section. However, there are few additional points here that need to be addressed:

1. Please justify the PP4 threshold used in coloc. In my opinion $PP4 > 0.5$ is a bit low (in that case one also should use H4/H3 ratio). But to my knowledge, it is more common to use $PP4 > 0.8$.
2. If one would like to link eQTLs with a specific putative enhancers, I think one also needs to fine-map the region at the minimum. This somewhat echos what other reviewers mentioned - I think that looking for the overlaps between variants and peaks in global, is very useful when assessing the cell type specific disease enrichment. However, making conclusions on individual loci is very error prone. In addition to the fine-mapping, at the minimum, I would remove the terms such as “NCM-specific cis-regulatory region” as it is still only a “peak” without experimental validation that it impacts OAS1 expression.

Minor comments:

I personally am not a fan of representing QTLs as bar plots. I would suggest to convert them to boxplots combined with dot plots as it is easier to see the error bars and how they overlap between genotypes. But I leave this decision to the authors.

Reviewer #6 (Remarks to the Author):

The authors have made a big effort to reply to all reviewers comments and performed extra analysis of localisation and TWAS to find putative causal genes using specific immune cell types gene expression

and atac-seq data.

I feel that reviewers concerns were adressed in this much improved paper. My only concern left is that, although the importance of the immune system cell types in Covid-19 is well established, it could be interesting to show that the studied cell types have an enrichment of signal for Covid-19. Using already existing atac-seq data for the cell types and a software like the one described in GARFIELD <https://www.nature.com/articles/s41588-018-0322-6> it would be easy to show in which cell types there is an enrichment of GWAS signal in atac-seq open chromatin sites, gene expression could be also used to show this.

As a minor comment:

- in line 105 "Here, we found that many colocalized severe COVID-19-risk variants in chromosome 21 were associated with reduced expression of the gene encoding interferon receptor 2 (IFNAR2)" it is not clear whether the many colocalized come from the different cell types or by testing different snps. If it comes from the different cell types I would put which ones rather than many, if it comes from testing snps in LD it is expected as coloc does not test a single variant but a region and it correlates the eqtl signal with the gwas signal, so it should give similar results for all the LD variants tested. So if it's the second case I would remove the comment unless you have tested independent variants.

Reviewer #3 (Remarks to the Author):

1. Most of the concerns that I raised in my original review have since been addressed and as such the study appears suitable for publication.

We are pleased that the reviewer is satisfied with our revision.

2. However, I first request that the authors make some version of the code they used to perform the analysis publicly available (on github or another repository) as opposed to available upon request. Having access to the exact code used is invaluable for understanding the steps performed for the analysis and reproducing the main findings.

As suggested by the reviewer, the codes are now publicly available on Github; <https://github.com/vijaybioinfo> (colocalization analysis, TWAS analysis and GWAS overlap analysis) or <https://github.com/ay-lab> (ATAC-seq and H3K27ac HiChIP).

Reviewer #4 (Remarks to the Author):

3. Schmiedel, Chandra, and Rocha et al. report potential genetic mechanisms mediating COVID-19 illness using an array of genomic data types from diverse immune cell types. In the interest of full disclosure, I previously reviewed a version of this manuscript at Nature Genetics. I do find the manuscript to be quite improved and the authors have made marked effort to address my previous concerns which I've outlined below along with any additional remaining concerns.

We thank the reviewer for the excellent suggestion that was incorporated in the previous revision.

4. Previously, my concern was with the rigor in which the “causative” variants were defined. The authors have addressed this in two primary ways – (1) They have incorporated ATAC-seq data and (2) They have, in some cases, delved further into the underlying mechanisms of the genetic association. This is particularly true with the anecdote shown in Figure 2d. This is very strong and I would say that it is almost certain that the authors have identified the causative allele. This is precisely the type and depth of analysis I was originally hoping to see. The remaining anecdotes now include additional data but do not rise to the same level of confidence as the one shown in Figure 2d. The anecdote in Figure 2e is less convincing due to the absence of a GATA motif overlapping the SNP of interest. It is nice to see that the GATA3 ChIP-seq signal lines up but the actual sequence change does not reside in or near a canonical GATA factor motif (as far as I can tell). Similarly, the anecdotes presented in Figure 3d and Figure 3g lack specific mechanisms and do not posit alteration of a specific motif. While I feel like this is the current standard in the field to nominate a putative causative noncoding variant, I also acknowledge that it may not be possible to find additional examples like the one presented in Figure 2d. In this case, I think the authors should at least mention this as a caveat and indicate that other cell types or cell states not assayed in this work could better explain the given associations.

We thank the reviewer for carefully reviewing the manuscript. For our most relevant findings (*OAS1*, *OAS3*, *DTX1* and *IL10RB*), we attempted to identify the potential underlying mechanism in the specific affected immune cell types. Unfortunately, this was not feasible in some cases, in part for the lack of data on the modulation of binding of transcription factors (TF) on sites of interest that may explain the observed effects of eQTLs on gene expression, especially in cases of immune cell types where convincing ChIP-seq data on a broad range of transcription factors is missing (in case of named eGenes in particular B cells and NK cells). We attempted to name candidate variants in **Supplementary Table 7**, predicted based on ENCODE TF binding data and made available via the HaploReg database.

In case of the predicted modulation of a GATA3 binding site at the site of the OAS3 enhancer (**Fig. 2e**), we agree with the reviewer that the discussed variant rs1298962 does not overlap the nearby canonical GATAA motif. However, the HaploReg database, relying on data from the study by Kheradpour and Kellis (Nucleic Acids Research 2013), suggested that the SNP is strongly affecting the binding of GATA. We found the canonical GATA motif located 10 nt upstream (located on the reverse strand as CTATT), and GATA3 binding to this region may thus be perturbed indirectly due to changes in the region neighboring the transcription factor motif as the SNP may affect enhancer syntax and the TF's binding affinity (Farley *et al.* Science 2015; Farley *et al.* PNAS 2016). This is the main reason, we considered a putative role of GATA3 for the regulation of OAS3 expression in naïve T cells, and added this detailed analysis to **Fig. 2e**. Further confidence was given to this conclusion by utilizing the fine-mapping method FINEMAP, which identified the OAS3 eQTL rs1298962 as a causal variant (**Supplementary Table 6**).

We have included these details and caveats in the revised manuscript:

“Notably, the OAS3 eQTL (rs1298962) was identified as a causal variant by fine-mapping (**Supplementary Table 6**)...”

“Of note, we also identified a canonical GATA motif 10 nucleotide upstream of rs1298962 (**Fig. 2e, bottom panel**), which led us to hypothesize that this variant may modulate the binding affinity of GATA3 either by perturbation of a submaximal recognition motif that affects enhancer syntax ("suboptimization") or by motif independent mechanisms⁴⁰⁻⁴².”

“However, COVID-19-risk variants may display stronger associations with gene expression in other immune cell types and activation conditions not examined in this work.”

5. Beyond identifying additional high-confidence anecdotes, I have the following minor comments:

The x axis in Figure 2a has no label. The y axis reads “Expression” but this is not sufficiently descriptive. The figure legend says TPM - please label as such. This also applies to the other figures of this type throughout the manuscript.

We thank the reviewer for pointing this out and adjusted the labels to “Expression (TPM)” across the manuscript.

6. In Table 1, the authors could add a column indicating their predicted causative variant for the relevant loci.

We thank the reviewer for this suggestion and agree this is a useful information for the reader. However, due to the complexity of identifying causal variants discussed above, we feel listing specific variants might be misleading as our different approaches may lead to false associations, in particular as they may differ for a given gene across various different immune cell types. For example, the colocalization and TWAS analysis merely predicts candidate eGenes in specific cell types but not specific functional variants. Similarly, the GWAS overlap analysis identifies the strongest eQTL for a given eGene in each affected cell type but this SNP may not constitute the causal variant. In addition, eGenes like *IL10RB* in NK cells (as indicated in **Fig. 3g**) may be affected by multiple variants affecting several enhancers. For a given region, fine-mapping may predict several candidates, but these potential causal variants lack extensive experimental validation. We would prefer to avoid calling any variant causal unless its function is validated experimentally.

We feel that the provided details and results from these methods in manuscript and supplementary materials will allow the reader to make conclusions. Details on potential causal variants can be found in **Supplementary Table 2** (eGenes and peak eQTLs from GWAS overlap analysis), **Supplementary Table 6** (fine-mapping of selected regions) and **Supplementary Table 7** (prediction of perturbed TF binding sites).

Reviewer #5 (Remarks to the Author):

7. In this paper, Schmiedel et al., used the DICE data to investigate how GWAS variants associated with the COVID-19 risk impact gene expression in different immune cell types. I agree what other reviewers have said, and believe that this is still largely a preliminary and descriptive analysis. The novelty of the study is limited, or should be better explained. As a substitute for reviewer 4, I believe that the comments suggested by a reviewer have been addressed. As suggested, the authors performed both coloc and TWAS which strengthened their conclusions, compared to simple overlaps. The authors also improved their method section. However, there are few additional points here that need to be addressed:

We agree with reviewer's comment on the initial version of manuscript and appreciate highlighting the improvements made with the revision.

8. Please justify the PP4 threshold used in coloc. In my opinion $PP4 > 0.5$ is a bit low (in that case one also should use H4/H3 ratio). But to my knowledge, it is more common to use $PP4 > 0.8$.

We thank the reviewer for this notion. Screening the literature, we found a wide range of thresholds being used to report colocalization results. We initially based our decision on the results from our sensitivity analyses (**Supplementary Fig. 3a**), and the suggestion that the conclusion of colocalization by a decision rule of $PP4 > 0.5$ is valid if prior beliefs are that $PP4$ is at least as likely as $PP3$ (Wallace PLOS Genetics 2020). In addition, we found several recent studies using a decision rule of $PP4 > 0.5$ (e.g., Huckins *et al.* Nat Genet 2019; GTEx Consortium Science 2020; Hu *et al.* Nat commun 2021; Jerber *et al.* Nat Genet 2021).

We agree with the reviewer that findings with $PP4 > 0.5$ are likely of lower confidence (Sun *et al.* Nature 2018). As suggested by the reviewer, we implemented the $PP4/PP3$ ratio ≥ 5 as an additional threshold for our analysis to enhance the confidence of our results from colocalization analysis (Huang *et al.* Nat Commun 2020; Li *et al.* Nat Commun 2019). Notably, this did not change the overall conclusions of our study.

Manuscript and **Supplementary Table 3** were adjusted accordingly:

“GWAS variants in 5 independent COVID-19-risk loci showed high posterior probability of colocalization ($PP4 > 0.5$ and $PP4/PP3$ ratio ≥ 5)^{15,18-21} with eQTLs associated with the expression of 9 eGenes (**Table 1**, **Fig. 1a**, **Supplementary Fig. 3a,b** and **Supplementary Table 3**).”

9. If one would like to link eQTLs with a specific putative enhancers, I think one also needs to fine-map the region at the minimum. This somewhat echos what other reviewers mentioned - I think that looking for the overlaps between variants and peaks in global, is very useful when assessing the cell type specific disease enrichment. However, making conclusions on individual loci is very error prone. In addition to the fine-mapping, at the minimum, I would remove the terms such as “NCM-specific cis-regulatory region” as it is still only a “peak” without experimental validation that it impacts OAS1 expression.

We thank the reviewer for this excellent suggestion. We have now performed fine-mapping of COVID-19-risk variants using FINEMAP tool (Benner *et al.* Bioinformatics 2016) and found that the majority of genetic variants previously prioritized by our approach using H3K27ac ChIP-seq and HiChIP assays and colocalization analysis are also identified as causal variants by fine-mapping (**Supplementary Table 6**). This includes the eQTL variants rs4767032 for *OAS1* in non-classical monocytes and rs1298962 for *OAS3* in naïve T cells that were discussed in detail in **Fig. 2**.

As suggested, we have modified the section on “NCM-specific cis-regulatory region” as below:

“Although the *OAS1* promoter did not directly overlap COVID-19-risk variants, we found several fine-mapped^{34,35} COVID-19-risk variants, associated with *OAS1* expression (eQTLs) (**Supplementary Table 6**), directly overlapped an intergenic transposase accessible and H3K27ac enriched peak region, located 20 kb away from *OAS1* promoter (**Fig. 2d**). This potential *cis*-regulatory region directly interacted with *OAS1* promoter in NCM (**Fig. 2d**), which suggested that perturbation of its activity by *OAS1* eQTLs is likely to explain their cell type-specific effects. The *OAS1* eQTLs in the NCM-specific H3K27ac peak region were predicted to disrupt the binding sites of several transcription factors (**Supplementary Table 7**).”

10. Minor comments:

I personally am not a fan of representing QTLs as bar plots. I would suggest to convert them to boxplots combined with dot plots as it is easier to see the error bars and how they overlap between genotypes. But I leave this decision to the authors.

We thank the reviewer for the suggestion allowing us to improve the presentation of the data. We adjusted the expression plots accordingly.

Reviewer #6 (Remarks to the Author):

11. The authors have made a big effort to reply to all reviewers comments and performed extra analysis of colocalisation and TWAS to find putative causal genes using specific immune cell types gene expression and atac-seq data. I feel that reviewers concerns were addressed in this much improved paper.

We thank the reviewer for supporting the value of our work.

12. My only concern left is that, although the importance of the immune system cell types in Covid-19 is well established, it could be interesting to show that the studied cell types have an enrichment of signal for Covid-19. Using already existing atac-seq data for the cell types and a software like the one described in GARFIELD <https://www.nature.com/articles/s41588-018-0322-6> it would be easy to show in which cell types there is an enrichment of GWAS signal in atac-seq open chromatin sites, gene expression could be also used to show this.

We thank the reviewer for carefully reviewing the revised manuscript. As suggested by the reviewer, we analyzed the ATAC-seq data from human immune cells (DICE) and chromatin accessibility data ('peaks') from cell types and tissues derived from ENCODE, GENCODE, and Roadmap Epigenomics using GARFIELD (Iotchkova *et al.* Nat Genet 2019). We identified enrichment of COVID-19 GWAS signals in regulatory regions of blood cells and immune cell types from DICE (**Supplementary Fig. 1b** and **Supplementary Table 1**).

The manuscript was modified accordingly:

“To predict cell types that are likely to be major contributors of the genetic risk of COVID-19, we first assessed enrichment of COVID-19-risk variants in *cis*-regulatory regions from a wide range of cell types and tissues using GARFIELD¹³ software. As expected, COVID-19-risk variants showed significant enrichment in chromatin accessibility regions (ATAC-seq and DNase-seq peaks) from blood cells and immune cell types, but little or no enrichment in other tissues and cell types (**Supplementary Fig. 1b** and **Supplementary Table 1**).”

13. As a minor comment:

- in line 105 "Here, we found that many colocalized severe COVID-19-risk variants in chromosome 21 were associated with reduced expression of the gene encoding interferon receptor 2 (IFNAR2)" it is not clear whether the many colocalized come from the different cell types or by testing different snps. If it comes from the different

cell types I would put which ones rather than many, if it comes from testing snps in LD it is expected as coloc does not test a single variant but a region and it correlates the eqtl signal with the gwas signal, so it should give similar results for all the LD variants tested. So if it's the second case I would remove the comment unless you have tested independent variants.

We thank the reviewer for pointing this out. Colocalization analysis identified the expression of *IFNAR2* to be affected in various immune cell types. We modified the manuscript accordingly:

"Here, we found that colocalized severe COVID-19-risk variants in chromosome 21 were associated with reduced expression of the gene encoding interferon receptor 2 (*IFNAR2*) in 12 of 13 immune cell types analyzed (**Fig. 2a**)."

We hope the revised manuscript is now suitable for publication in Nature Communications.

REVIEWERS' COMMENTS

Reviewer #5 (Remarks to the Author):

The authors adequately addressed all my concerns.

Minor comment:

line 155: "...a causal variant by fine-mapping..." I would suggest to add likely or putative causal.

Reviewer #6 (Remarks to the Author):

The authors have adressed all my concerns and I think the manuscript is ready for publication now.

Reviewer #5 (Remarks to the Author):

The authors adequately addressed all my concerns.

Minor comment: line 155: "...a causal variant by fine-mapping..." I would suggest to add likely or putative causal.

We thank the reviewer for this excellent suggestion. We have adjusted the text accordingly.

Reviewer #6 (Remarks to the Author):

The authors have adressed all my concerns and I think the manuscript is ready for publication now.

We thank the reviewer for suggestions and comments to improve the initial version of manuscript.